# A Tail of Two Clumps

**Alejandro Cristian Raga** [1], **Jorge Cantó** [2], **Antonio Castellanos-Ramírez** [2], **Jorge Ivan Castorena** [1], **Liliana Hernández-Martínez** [3], **Ary Rodríguez-González** [1] and **Pedro Rivera-Ortíz** [4,*]

[1] Instituto de Ciencias Nucleares, Universidad Nacional Autónoma de México, Ap. 70-543, Mexico City 04510, Mexico; raga@nucleares.unam.mx (A.C.R.); jicastorena@correo.nucleares.unam.mx (J.I.C.); ary@nucleares.unam.mx (A.R.-G.)

[2] Instituto de Astronomía, Universidad Nacional Autónoma de México, Ap. 70-658, Mexico City 04510, Mexico; jcanto@astro.unam.mx (J.C.); acastellanos@astro.unam.mx (A.C.-R.)

[3] Facultad de Ciencias, Universidad Nacional Autónoma de México, Ap. 70-543, Mexico City 04510, Mexico; lilihe.mtz@gmail.com

[4] CNRS, IPAG, University Grenoble Alpes, 38000 Grenoble, France

* Correspondence: pedr10@hotmail.com

**Abstract:** We present two axisymmetric simulations of a high velocity clump in a photoionized region: one for the case of a uniform, low density environment and a second one for the case of a clump first traveling within a high density medium and then emerging into a low density environment. We show that the second scenario results in the production of an axial tail of dense material with a linear velocity vs. position ramp (with zero velocity at the high/low density environment transition). This material comes from a confined bow shock (produced by the clump when it was within the dense cloud) that emerges into the low environmental density region.

**Keywords:** planetary nebulae; nebulae around massive stars; shaping mechanisms

## 1. Introduction

In different astrophysical objects, we find "fast clump" or "interstellar bullet" flows with a long tail of material with linearly decreasing velocities towards the outflow source. For example, this kind of flow is found in planetary nebulae (PNe) or in proto-planetary nebulae (PPNe) (see, e.g., [1,2] and the review of [3]).

Modelling high speed "bullet" flows in terms of winds, jets and clumps ejected from the central source has been presented by [4,5]. In a recent series of papers [6–8], we showed that an ejection pulse with either of the following bulleted points result in the formation of a leading bow shock followed by a tail of material with an approximately linear velocity growth from the source position out to the emitting clump:

- A modulated ejection velocity peak (we studied the case of a parabolic and a Gaussian time-dependence of the ejection velocity);
- A non-top hat initial velocity cross section.

In the present paper, we show that a "clump flow" (with an initially spherical, dense clump with uniform density and velocity within the clump) can also produce a linear velocity vs. position tail. Such a tail is produced if the clump first travels within a dense region and then emerges into a much lower density environment.

In order to show that this is the case, we have computed axisymmetric numerical simulations (with the setup described in Section 2) of a dense, high velocity clump traveling in a low density environment and of a clump first traveling in a high density environment and then emerging into a low density medium. The results of these simulations are described in Section 3, and predictions of Hα emission maps and position-velocity (PV) diagrams are shown in Section 4. The results are summarized in Section 5.

## 2. The Numerical Simulations

In order to present a first study of the problem, we have used a simple setup: We consider a domain in which H is always fully ionized and have used a parametrized cooling function (in particular, the one of [9]) and impose a minimum temperature of $10^4$ K in all grid cells at all times. This setup is a "poor man's approach" for simulating flow within a photoionized region.

We use the adaptive grid "yguazú" code (see [7]) to integrate the axisymmetric gasdynamic equations (with the cooling function described above) in a $(10, 2.5) \times 10^{17}$ cm (axial $\times$ radial, "$x$, $r$") domain. A four-level, binary adaptive grid with a maximum number of $2048 \times 512$ (axial $\times$ radial) is used.

The initial configuration has a dense sphere of radius $R_0 = 10^{16}$ cm and atom+ion number density $n_0 = 10^4$ cm$^{-3}$ centered at $x_0 = 1.5 \times 10^{17}$ cm. The rest of the domain is filled with a uniform environment that flows along the $x$-axis at a velocity of $v_a = 300$ km s$^{-1}$ (with an inflow boundary on the left and an outflow boundary condition on the right). The initial domain has a uniform temperature of $10^4$ K.

We run the following two simulations:

A.    With an environment that is injected into the domain at all times with the $n_a = 10$ cm$^{-3}$ density of the initial condition;
B.    With an initial environment of density $n_c = 10^4$ cm$^{-3}$ and for $t > 0$ an injection with $n_a = 10$ cm$^{-3}$.

Model B is intended to model a clump that first travels within a dense circumstellar region of density $n_c$ and then emerges into a much lower density outer envelope of density $n_a$.

From the results of the numerical integrations, we compute the recombination cascade H$\alpha$ emission coefficient. With this emission coefficient, we calculate emission maps and position-velocity (PV) diagrams for a narrow spectrograph slit aligned with the projected symmetry axis.

## 3. Results

Figure 1 shows a time-sequence of the density structure obtained from model A. In this model, the highly overpressured cloud expands, and the streaming environment is deflected in a bow shock.

Figure 2 shows a similar time-sequence for model B. In the $t = 200$ yr frame, we observe that the cloud is confined by the streaming, dense environment. At $t = 600$ yr, the cloud has emerged into the low density, streaming environment region and has started to expand. By the end of the displayed time sequence (at $t = 2200$ yr), the cloud has expanded, possessing a similar size to the cloud (at 2200 yr) of model A.

A notable difference between model A and B (Figures 1 and 2, respectively) is that the region of the wake behind the cloud is much denser in model B. This dense, axial region is filled with materials belonging to the bow shock that was developed at early times when the clump traveled through the dense region of the environment (see the top frame of Figure 2), which then emerges out of the dense region, trailing behind the cloud.

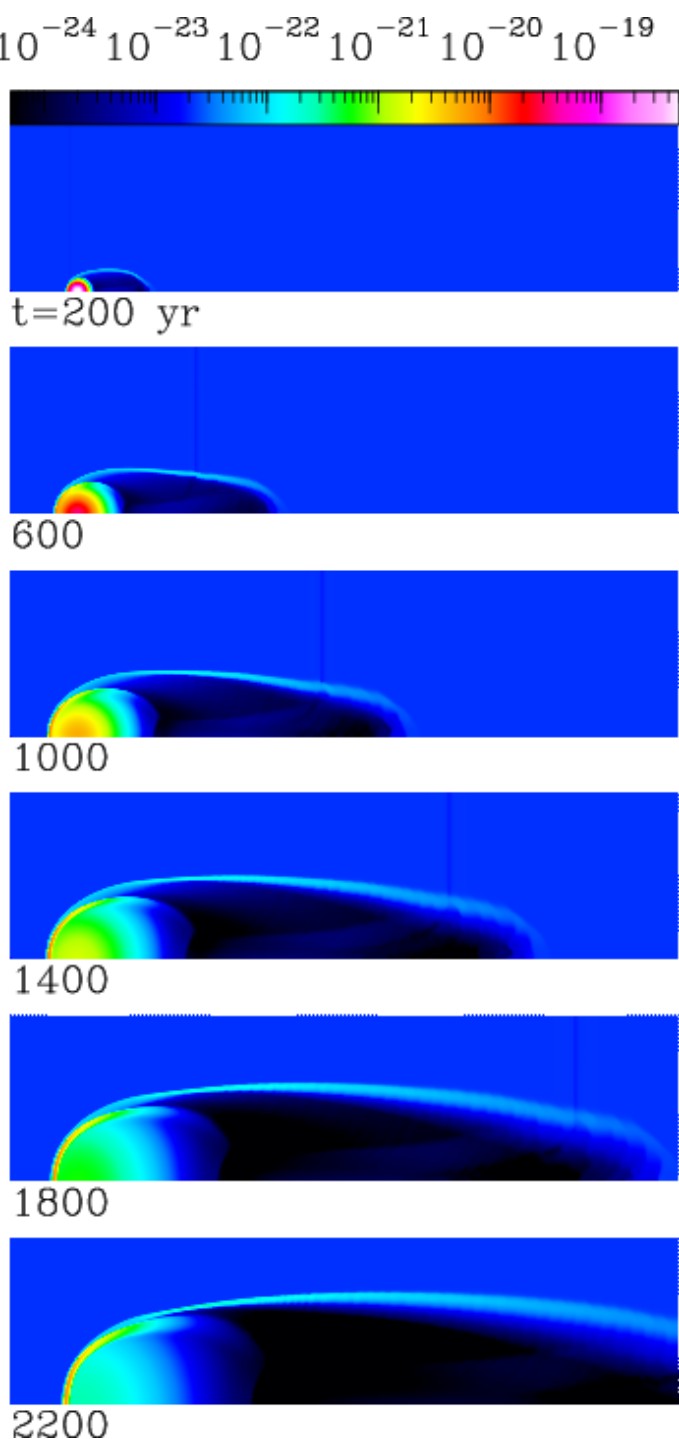

**Figure 1.** Density stratifications for different times of the model A evolution of a dense clump in a uniform streaming environment. The domain has an axial extent of $10^{18}$ cm, and the densities are shown with the logarithmic scale provided (in g cm$^{-3}$) by the top bar. The ion+atom number density can be obtained approximately by multiplying the mass density (shown in the plots) by a factor of $4.6 \times 10^{23}$ (as 0.9 H and 0.1 He fractional abundances are assumed). As H and He are fully ionized, the electron density will have values similar to the ion+atom number density.

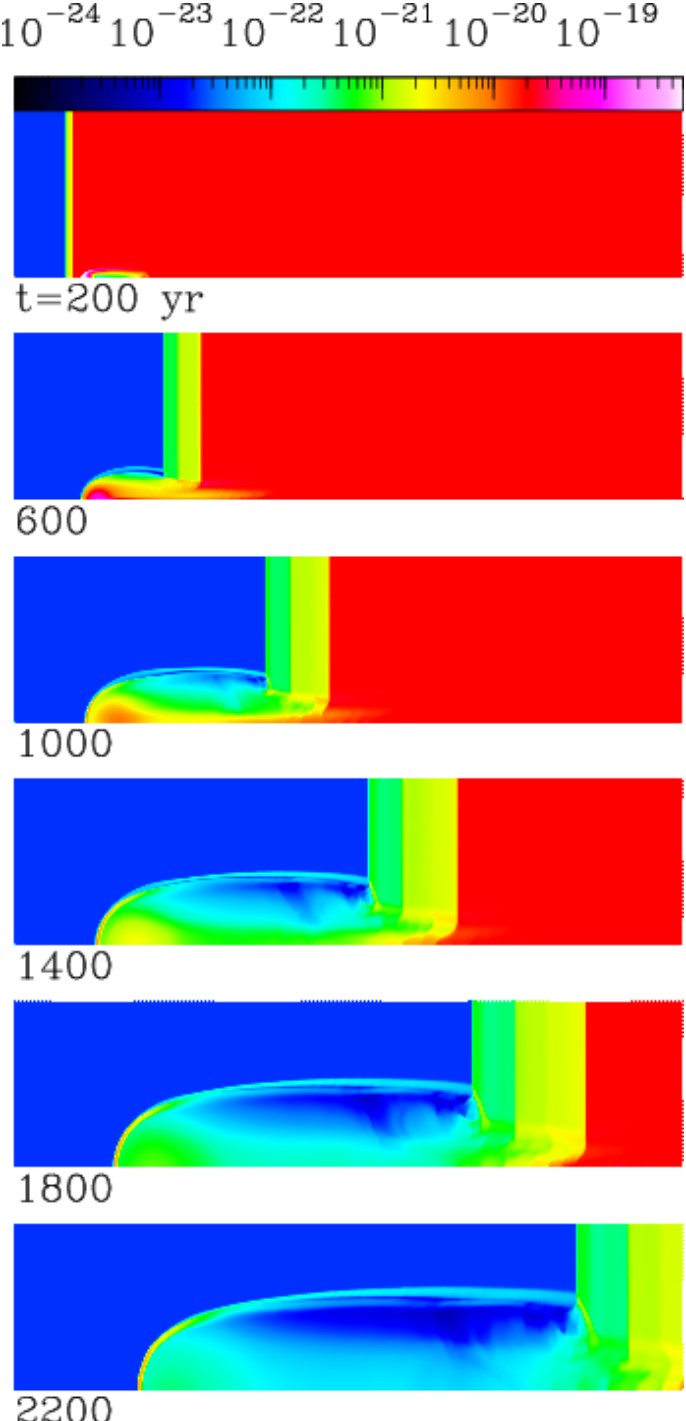

**Figure 2.** The same as Figure 1 but for model B of a clump first proceeding through a dense medium and then emerging into a low density environment.

## 4. Intensity Maps and PV Diagrams

From density and temperature stratifications at $t = 2200$ yr (bottom frames of Figures 1 and 2), we have computed H$\alpha$ intensity maps for models A and B, assuming that the flow axis lies on the plane of the sky. The resulting maps are shown in Figure 3.

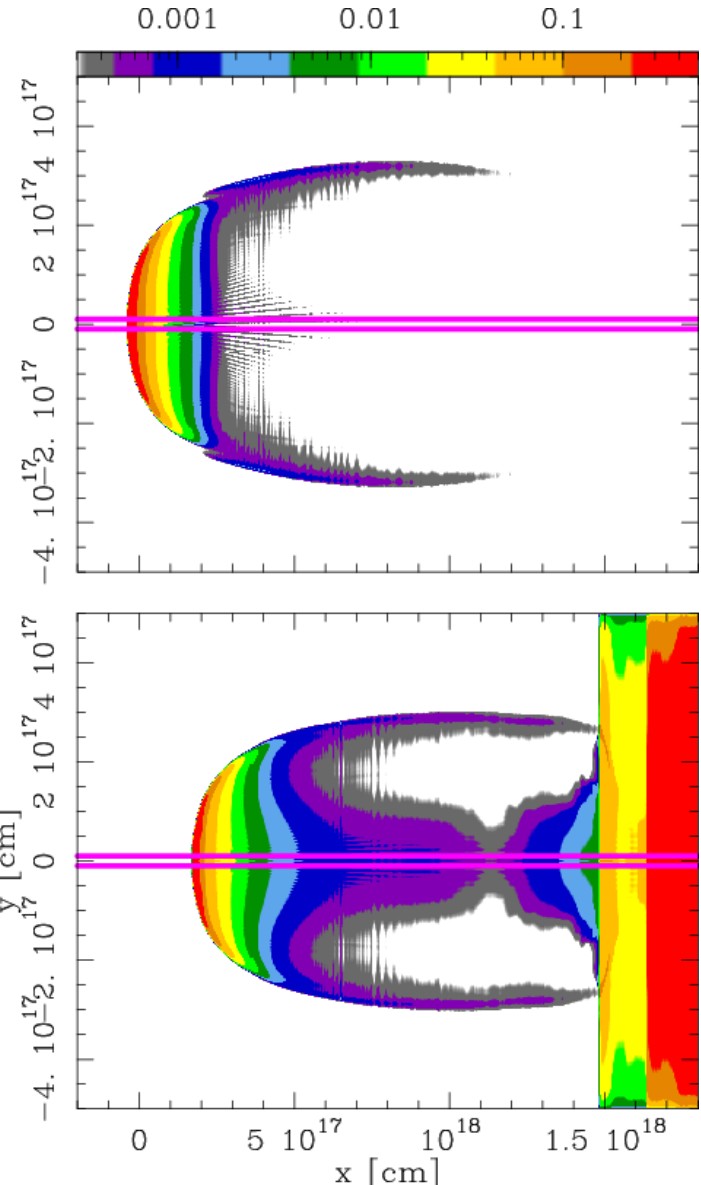

**Figure 3.** Hα intensity maps produced by models A (**top**) and B (**bottom**) for a $t = 2200$ yr integration time. The maps are calculated assuming that the flow lies on the plane of the sky. The emission (normalized to the peak emission at the tip of the bow shock) is given with the logarithmic color scale given by the top bar. The two purple horizontal lines show the spatial coverage of the long slit used to calculate the PV diagrams of Figure 4.

The Hα emission of model A only shows the bow shock. The map of model B shows a somewhat smaller bow shock. This map also shows emission from the dense environment (which has still not entirely left the computational domain, see Figure 4) and the axial emission of the material dragged out by the confined bow shock (within the dense environment region) into the low density environment region.

For the narrow spectrograph slits shown on Hα intensity maps of Figure 3, we have computed Hα PV diagrams, which are shown in Figure 4 (for an assumed angle of 30° towards the observed between the flow axis and the plane of the sky). The PV diagrams have been computed by placing the origin of the radial velocity axis at the projected velocity of the streaming environment so that the diagrams correspond to an observer at rest with respect to the environmental material.

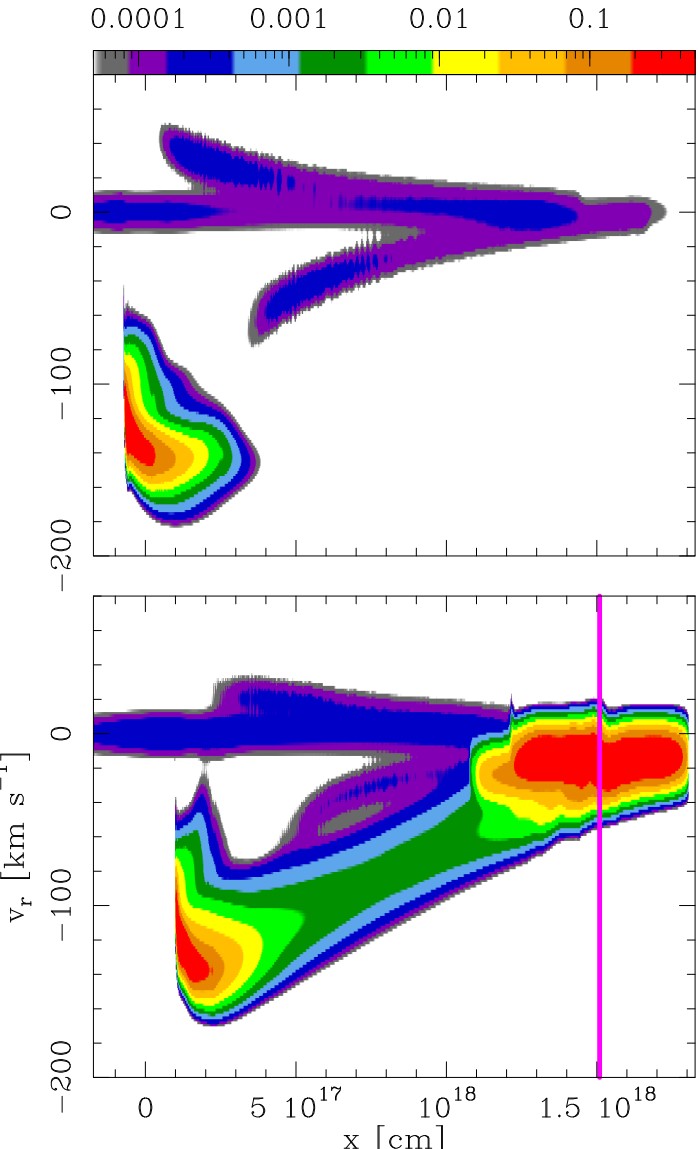

**Figure 4.** Hα PV maps produced by models A (**top**) and B (**bottom**) for a $t = 2200$ yr integration time. The maps are calculated for the spectrograph slit positions shown in Figure 3, and it is assumed that the flow points towards the observer at a 30° angle from the plane of the sky. The emission (normalized to the peak emission at the tip of the bow shock) is given with the logarithmic color scale provided by the top bar.

In Figure 4, we observe that the PV diagram of model A is dominated by the leading bow shock. The PV diagram of model B also shows the strong, zero radial velocity emission of the dense region of the environment, and an approximately linear ramp of emission joining the bow shock to the position at which the cloud emerged into the low density region (shown with a vertical purple line in the bottom frame of Figure 4). This emission comes from the on-axis region of dragged out dense environment observed in model B's density frames of Figure 2 (and also observed in the corresponding Hα map of Figure 3).

The PV diagrams of Figure 4 have of course been calculated for an arbitrary choice for the orientation angle (30° between the outflow axis and the plane of the sky, see above). However, the same qualitative features (including the linear ramp in the model B PV diagram) are observed for orientation angles of ∼15→ 60°.

## 5. Discussion

We have presented a model for the production of clumps with a linear velocity vs. position tail. In order to produce the tail, a fast, high density clump initially travels through a high density region and then emerges into a low density medium. In the context of PNe, this could correspond to a fast, collimated ejection that first travels within a dense, inner region of the nebula and then emerges into a low density halo.

We have presented two simulations of photoionized bullets:

- A. Traveling within a uniform, low density medium;
- B. Initially traveling in a high density region and then emerging into a low density medium.

While model A produces a standard bow shock flow, with bow-shaped intensity maps and "comma shaped" PV diagrams (to use the description of [4]), model B also shows a dense axial structure corresponding to material in the "confined bow shock" (produced when the bullet travels within the high density region) that emerges into the low density medium. As it emerges from the dense region, the released bow shock material moves approximately ballistically. A "velocity sorting" process occurs, with the fast bow shock region close to the clump head following the clump and the slower moving "bow shock wing" material trailing behind.

In this manner, model B has a "tail" of axially concentrated material. This tail shows up in PV diagrams as a linear ramp increasing from zero radial velocity at the point of emergence from the dense cloud up to the clump velocity (at the position of the clump).

This kind of flow clearly has applications to high velocity clumps in PNe followed by linear velocity ramps (e.g., in the classical example of OH 231.8 + 4.2, see [1], and possibly also in multi-bullet flows in star formation regions (see, e.g., [10] and references therein). Clearly, detailed models of specific objects will be necessary to observe if our model is indeed appropriate.

**Author Contributions:** The first author (A.C.R.) has contributed approximately 30%, and the other co-authors (J.C., A.C.-R., J.I.C., L.H.-M., A.R.-G., P.R.-O.) approximately 10% each of the reported work. All authors have read and agreed to the published version of the manuscript.

**Funding:** This research was funded by the DGAPA (UNAM) grant number IG100422.

**Institutional Review Board Statement:** Not applicable.

**Informed Consent Statement:** Not applicable.

**Data Availability Statement:** Not applicable.

**Acknowledgments:** A.Cas.R. acknowledges support from a CONACyT postdoctoral fellowship.

**Conflicts of Interest:** The funders had no role in the design of the study; in the collection, analyses, or interpretation of data; in the writing of the manuscript, or in the decision to publish the results.

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
