# Peer review of "A Tail of Two Clumps"

_galaxies, doi:10.3390/galaxies10010019_

Round 1
Reviewer 1 Report
This short paper is a very valuable contribution to the literature on
high velocity emission in proto- / planetary nebulae and extends the
work of this group. The authors have revealed a very relevant situation
where a high velocity clump emerges from the higher density shell into a
lower density outer shell/halo. The paper should be published and I have
just a few minor points.
Abstract
l.3 to read .. into a low density ...
Sect. 2
para 2, last line
Do the authors mean 2024x512 cells ?
Figures
All the figures refer to models M1 and M2 but in Sect. 2 they are
referred to as A and B. This should be made consistent.
The density is given in g cm-3 but it would be very useful to
state the conversion to (the easily measurable) electron number
density since this depends on the ionization of the gas (and the
fraction of elements heavier than H).
Fig.3
It would be useful to state in the caption that the parallel lines
show the extent of the slit for which profiles are shown in Fig. 4.
Sect. 4
Para 3
Why is an angle of 30 deg between flow axis and plane of sky chosen?
If this is just for illustration it should be stated, as there can be
nothing fundamental about this choice of angle.
Sect. 5
It would be very useful to given examples of PN/p-PN in which the
bullet situation B occurs, in order to inspire observers to study
them more closely. If the origin of the linear velocity ramp behind
a bullet in situation B occurs at the denser/less dense shell interface,
it produces clearly observable evidence for this model from radial
velocity data.
last line
... of our model -> if our model ...
Author Response
we have carried out the corrections suggested by the referee as follows:
- abstract: we have corrected the text in l,3
- sect. 2: 2048x512 is correct!
- we have corrected the notable mistaken identification of the
models in the figure captions (the models are now uniformly referred
to as A and B)
- we now give the conversion between mass density (shown in figures
1 and 2) and ion+atom number density in the caption of figure 1
- in the caption of figure 3, we now explain that the two horizontal
lines show the spectrograph slit used for the PV diagrams of figure 4
- section 3, paragraph 4: indeed there is nothing fundamental about
this choice of orientation angle. We have now added a small text
at the end of section 3 discussing for what range of orientation angles
similar PV diagrams are obtained
- we now give as an example of clumps with "Hubble velocity law
tails" the paper of Alcolea et al. (2001) in the last paragraph
of the paper
- last line is now corrected
Reviewer 2 Report
A few minor comments. The two models are called A and B in the text but M1 and M2 in the figures. In Figure 4, the label of the vertical axis cut off so that the power '-1' is missing from the units. The references have numbers '9' and '10' attached for unclear reason
Author Response
- the models are now uniformly referred to as "A" and "B"!
- -we have corrected figure 4 so that the labels are now fully
visible